# DAFCNN: A Dual-Channel Feature Extraction and Attention Feature Fusion Convolution Neural Network for SAR Image and MS Image Fusion

**Jiahao Luo [1], Fang Zhou [1,\*], Jun Yang [2] and Mengdao Xing [1,3]**

1   School of Computer Science and Information Engineering, Hefei University of Technology, Hefei 230009, China
2   College of Geomatics, Xi'an University of Science and Technology, Xi'an 710071, China
3   National Key Laboratory of Radar Signal Processing, Xidian University, Xi'an 710071, China
\*   Correspondence: zhoufang@hfut.edu.cn

**Abstract:** In the field of image fusion, spatial detail blurring and color distortion appear in synthetic aperture radar (SAR) images and multispectral (MS) during the traditional fusion process due to the difference in sensor imaging mechanisms. To solve this problem, this paper proposes a fusion method for SAR images and MS images based on a convolutional neural network. In order to make use of the spatial information and different scale feature information of high-resolution SAR image, a dual-channel feature extraction module is constructed to obtain a SAR image feature map. In addition, different from the common direct addition strategy, an attention-based feature fusion module is designed to achieve spectral fidelity of the fused images. In order to obtain better spectral and spatial retention ability of the network, an unsupervised joint loss function is designed to train the network. In this paper, the Sentinel 1 SAR images and Landsat 8 MS images are used as datasets for experiments. The experimental results show that the proposed algorithm has better performance in quantitative and visual representation when compared with traditional fusion methods and deep learning algorithms.

**Keywords:** SAR and MS images; image fusion; convolutional neural network (CNN)

## 1. Introduction

With the rapid development of remote sensing technology, the types of imaging sensors have become more and more diverse. In terms of remote sensing systems, high spectral resolution and spatial resolution often cannot be obtained simultaneously [1,2]. The synthetic aperture radar is an active microwave sensor whose imaging process does not depend on signal from the sun. This allows SAR systems to operate all day and under all weather circumstances. However, it also leads to SAR images that do not contain spectral information [3,4]. Multispectral sensors are passive and rely mainly on the ability of the ground to reflect light, which can visually respond to the color and texture of the object; however, they have a low resolution [5]. Therefore, fusing MS images and SAR images to generate high-quality fused images with richer spatial information and spectral feature greatly improves the structural information of the source images and can obtain the hidden information in SAR images, which enhances the image interpretation capability. It facilitates the subsequent tasks such as urban land cover, terrain classification, and road detection.

Currently, image fusion algorithms can be divided into four main categories: component substitution (CS), multiresolution analysis (MRA), hybrid methods, and deep learning-based methods.

The CS method projects the low-spatial-resolution MS images to other spaces, thus separating the spectral and spatial information, and replaces the spatial components with high-spatial-resolution SAR images by histogram matching. Finally, the MS images are

inverted back to the original space to obtain the fusion results. Methods that function in this manner include: IHS transform [6–9], principal component analysis (PCA) [10–12], Brovery transform (BT) [13], Gram–Schmidt (GS) transformation [14], etc.

The MRA method decomposes the source image into different scale spaces using multiscale transformation methods first, such as Laplacian pyramid transform [15,16], wavelet transform [17–20], contourlet transform [21,22] and curvelet transform [23,24], non-subsampled contourlet transform (NSCT) [25–27], non-subsampled shearlet transform (NSST) [28]. Then, the image is fused in different scale space. Finally, the fusion result is obtained by inverse transformation.

Hybrid methods [29–31] combine the advantages of CS and MRA methods. The PCA transform can obtain higher spatial resolution, but produces more severe spectral distortion. On the other hand, wavelet transform is able to retain spatial information, but the result lacks high spatial resolution. YAN et al. [30] proposed a fusion technique based on additive wavelet decomposition and PCA transform to achieve remote sensing image fusion. Zhao et al. [31] proposed a fusion method based on IHS and NSCT transform, which effectively achieved unmanned aerial vehicle (UAV) panchromatic and hyperspectral image fusion.

In recent years, deep learning has produced excellent results in computer vision, natural language processing, and image processing due to its powerful feature representation [32]. Image fusion is an important branch of image processing, and researchers have proposed many deep learning-based image fusion methods [33–38]. K. Ram Prabhakar et al. [33] proposed the DeepFUSE network to implement extreme exposure image fusion. The network consists of three components: a feature extraction layer, a fusion layer, and a reconstruction layer. The feature extraction network extracts similar features from different source images through weight sharing and fuses them in the fusion layer to obtain the fusion results. On this basis, Li [34] proposed the DenseFuse method for the fusion of infrared and visible images. The encoded network obtained better fusion results by introducing dense linking blocks to link the features of each layer with other layers to get richer features. Yuan [35] proposed a multiscale and multidepth convolutional neural network (MSDCNN) which obtained good results in the pan-sharpening task. Yang et al. [37] proposed a new progressive cascade deep residual network with two residual sub-networks for the pan-sharpening task and obtained distortion-free fusion results. He [38] proposed a convolutional neural network-based method for arbitrary resolution traditional hyperspectral (ARHS-CNN) to achieve arbitrary resolution hyperspectral image fusion. Saxena [39] proposed a pan-sharpening method based on the multistage multichannel spectral graph wavelet transform and a convolutional neural network (SGWT-PNN).

Among the above four types of methods, CS methods have lower computational complexity, but these methods are highly dependent on the correlation between images. Owing to the large differences between SAR images and MS images, the fused images of SAR and MS images can show severe spectral distortion. MRA fusion methods can achieve fusion in different frequency spaces as a way to obtain better performance. However, MRA methods are extremely time consuming and heavily rely on decomposition methods and fusion rules. It is still difficult to choose the appropriate decomposition methods and fusion rules for different images. Deep learning methods can dispose of the high level of manual involvement, enabling the network to learn autonomously. However, most networks simply sum the feature maps without considering the connections between feature map channels. It is equally difficult to design a suitable network structure for a specific source image fusion task.

To overcome the above-mentioned limitations, this paper proposes a dual-channel feature extraction and attention fusion convolutional neural network (DAFCNN) to achieve SAR image and MS image fusion. In order to extract the features of high-resolution SAR images more effectively, we design a dual-channel feature extraction module to extract the spatial feature information of high-resolution SAR images at different levels. In addition,

an attention-based feature fusion (AFF) module is designed to fuse features from SAR image and MS image. The AFF module is different from the common direct summation fusion strategy, which fully considers the relationship between feature map bands and preserves the spectral characteristics of MS images. To train our network more effectively, an unsupervised joint loss function is proposed to constrain the network training. The experimental results in Section 4 show that the proposed method has excellent performance in the task of fusing SAR images with MS images.

The rest of this paper is organized as follows: Section 2 presents the related work, Section 3 describes the method proposed in this paper, Section 4 provides the experimental results and compares them with other methods, and finally, Section 5 makes a conclusion.

## 2. Related Work

### 2.1. Residual Block Structure

He et al. [40] proposed a residual learning network structure that overcomes the problem of vanishing network gradients while deepening the network. A residual block is shown in Figure 1. Formally, the residual block is represented as

$$y = F(x, \{W_i\}) + x \tag{1}$$

where $x$ and $y$ are the input and output vectors of the residual block. The function $F(x, \{W_i\})$ denotes the residual mapping to be learned. As in Figure 1, the $F(x)$ is represented as

$$F(x) = W_2\delta(W_1 x) \tag{2}$$

where $W$ is the weight of convolutional layer and $\delta$ is the ReLU activation function.

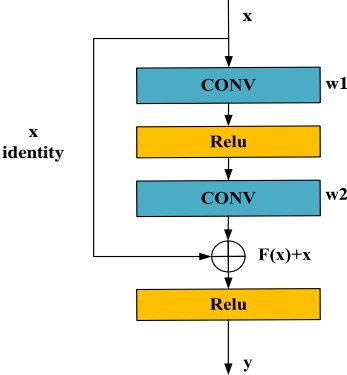

**Figure 1.** Residual block structure.

### 2.2. Squeeze-and-Excitation Networks

Hu et al. [41] proposed the channel attention module squeeze-and-excitation network (SENet), as shown in Figure 2. SENet uses global information to explicitly model dynamic, nonlinear dependencies between channels, which can simplify the learning process, suppress useless information, and significantly enhance the representation capability of the network. Furthermore, SENet is easy to integrate into other networks as an attention module. SENet includes a squeezing module and an excitation module. The squeeze module extracts the spatial information by global average pooling, which can compress the information without increasing the time and space complexity. The squeezing operation $F_{sq}(\cdot)$ flow equation is represented as

$$z_c = F_{sq}(U_c) = \frac{1}{H \times W} \sum_{i=1}^{H} \sum_{j=1}^{W} U_c(i, j) \tag{3}$$

where $U$ is the input feature map of the squeezing operation; $H$, $W$, $C$ represent the height, width, and channel of the feature map; and $z$ is the output feature map of the squeezing operation.

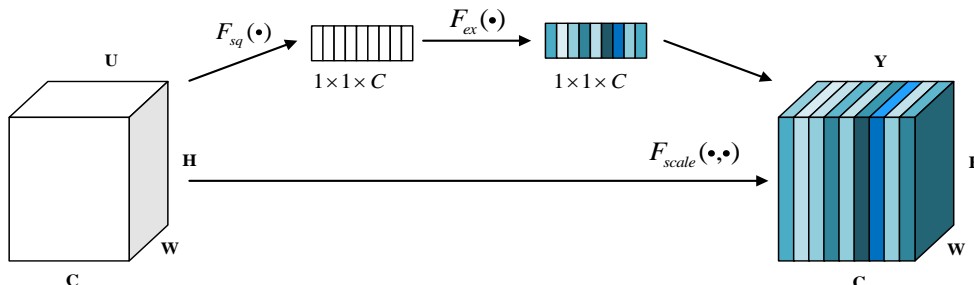

**Figure 2.** SENet.

To take advantage of the global information obtained from the squeeze operation, the excitation operation is then used to fully capture the channel correlation. The excitation operation $F_{ex}(\cdot)$ uses the sigmoid activation gate mechanism and the process equation is denoted as

$$s = F_{ex}(z, D) = \sigma(g(z, D)) = \sigma(D_2 \delta(D_1 z)) \tag{4}$$

where $s$ is the output feature map of the excitation operation, $\delta$ denotes the ReLU activation function, $\sigma$ denotes the sigmoid activation function, and $D_1, D_2$ denotes the fully connected operation. The output $Y$ of the final network is denoted as

$$Y = F_{scale}(U_c, s_c) = U_c s_c \tag{5}$$

where $F_{scale}(U_c, s_c)$ represents the input feature map $U$ multiplied by the scaling factor $s$ by channel.

### 2.3. Structural Similarity Index

Wang et al. [42] proposed an structural similarity evaluation metric (SSIM) to measure the similarity between two images. SSIM compares the differences between images in terms of luminance, contrast, and structure, respectively. The defined functions of luminance, contrast, and structure are as follows:

$$l(R, F) = \frac{2\mu_R \mu_F + C_1}{\mu_R^2 + \mu_F^2 + C_1} \tag{6}$$

$$c(R, F) = \frac{2\sigma_R \sigma_F + C_2}{\sigma_R^2 + \sigma_F^2 + C_2} \tag{7}$$

$$s(R, F) = \frac{\sigma_{RF} + C_3}{\mu_R \mu_F + C_3} \tag{8}$$

where $\mu_R, \mu_F, \sigma_R^2, \sigma_F^2, \sigma_{RF}$ denote the mean, variance of image and covariance between the two images, respectively. The function of $C_i (i = 1, 2, 3)$ is to avoid instability when the denominator of $l, c, s$ approaches 0. $C_i$ can be calculated as $C_1 = (K_1 L)^2, C_2 = (K_2 L)^2$, $C_3 = C_2/2$, usually set $K_1 = 0.01, K_2 = 0.03$. $L$ is the dynamic range of the image pixel values. The above three metrics are combined to form SSIM as

$$SSIM = [l(R, F)]^{\alpha} [c(R, F)]^{\beta} [s(R, F)]^{\gamma} \tag{9}$$

### 3. The Proposed Method

In this section, the structure of DAFCNN is described in detail, which consists of three modules: the spatial feature extraction branch, the spectral preservation branch, and the feature fusion module based on the attention mechanism. This section introduces the

specifics of these three modules at first, and then the designed loss function is introduced. The complete structure of the network is shown in Figure 3.

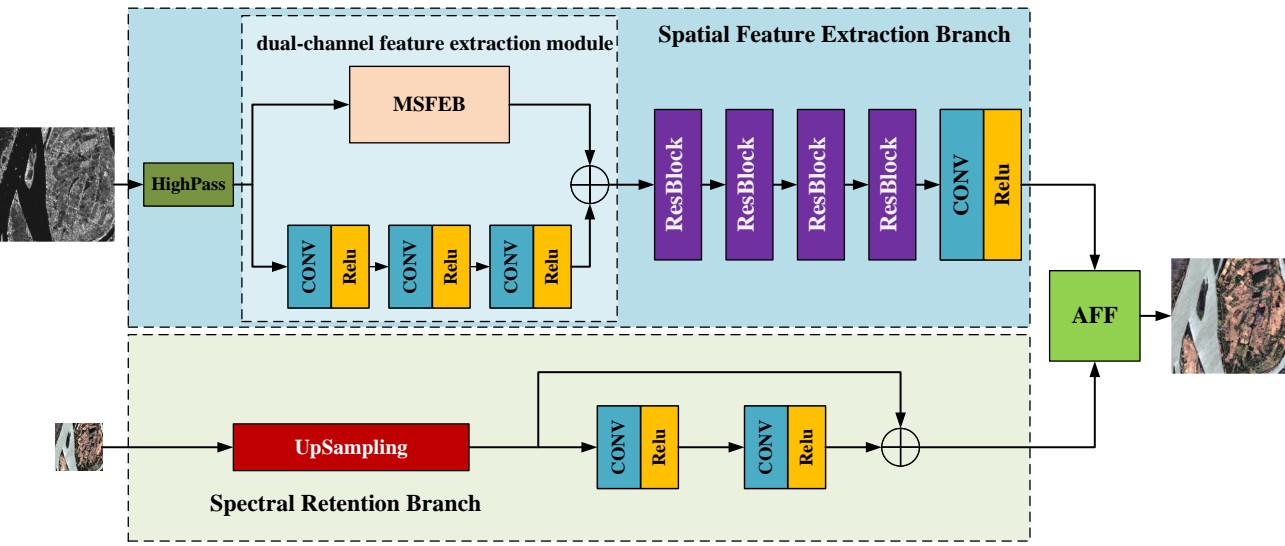

**Figure 3.** Detailed structure diagram of DAFCNN.

### 3.1. Spatial Feature Extraction Branch

The spatial feature extraction branch consists of a two-channel feature extraction network and four residual blocks. In order to efficiently and completely obtain the objects' information of various sizes in SAR images, a dual-channel feature extraction module is designed to extract the object features in SAR images. The dual-channel feature extraction module has two independent modules: a three layers basic convolutional module and a multi-scale feature extraction block (MSFEB). The three-layer structure is used to extract shallow features in SAR images. The MSFEB can extract intermediate and high-level features of SAR images, and its structure is shown in Figure 4a.

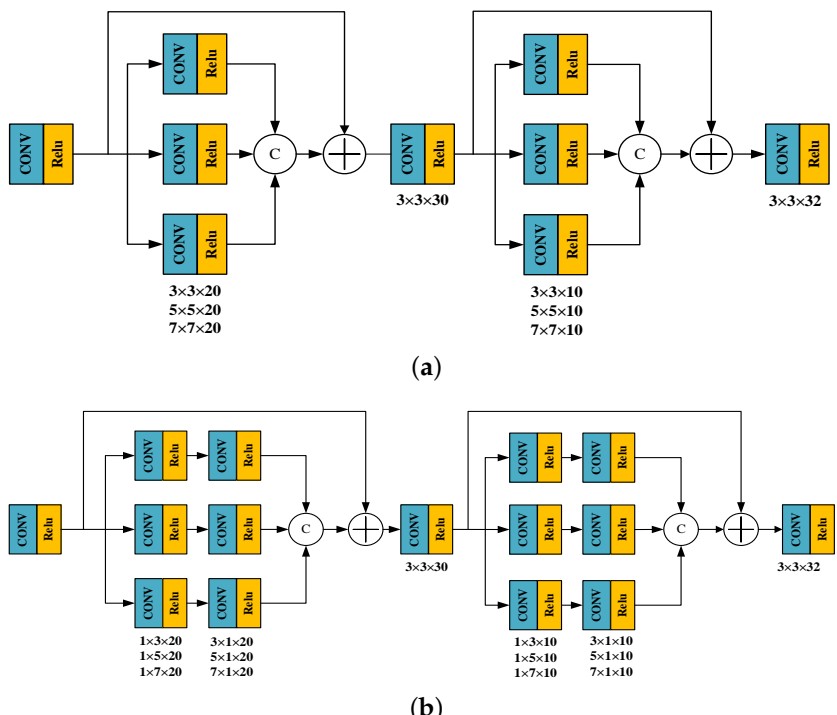

**Figure 4.** Structure of MSFEB. (**a**) General MSFEB, (**b**) improved MSFEB.

In general, large convolutional kernels are used to extract features of large targets, and small convolutional kernels are used to extract features of small targets. MSFEB uses convolutional layers with kernel sizes of $3 \times 3$, $5 \times 5$, and $7 \times 7$ to extract features at different scales, and then links the feature maps extracted from different kernels by channel. The MSFEB also introduces skip connections, which can make fuller use of the spatial information in SAR images and avoid information loss due to deeper network. In order to speed up the network training but not affect the network feature extraction capability, we replace the $3 \times 3$, $5 \times 5$, and $7 \times 7$ convolutional kernels with $1 \times 3$ and $3 \times 1$, $1 \times 5$ and $5 \times 1$, and $1 \times 7$ and $7 \times 1$ convolutional kernels. The improved MSFEB structure is shown in Figure 4b.

The following figure shows the feature maps extracted by MSEFB and the basic three-layer module (five groups selected from each feature map). As shown in Figure 5, it can be seen that MSEFB can extract finer ground texture details, whereas the basic three-layer module extracts rough structures.

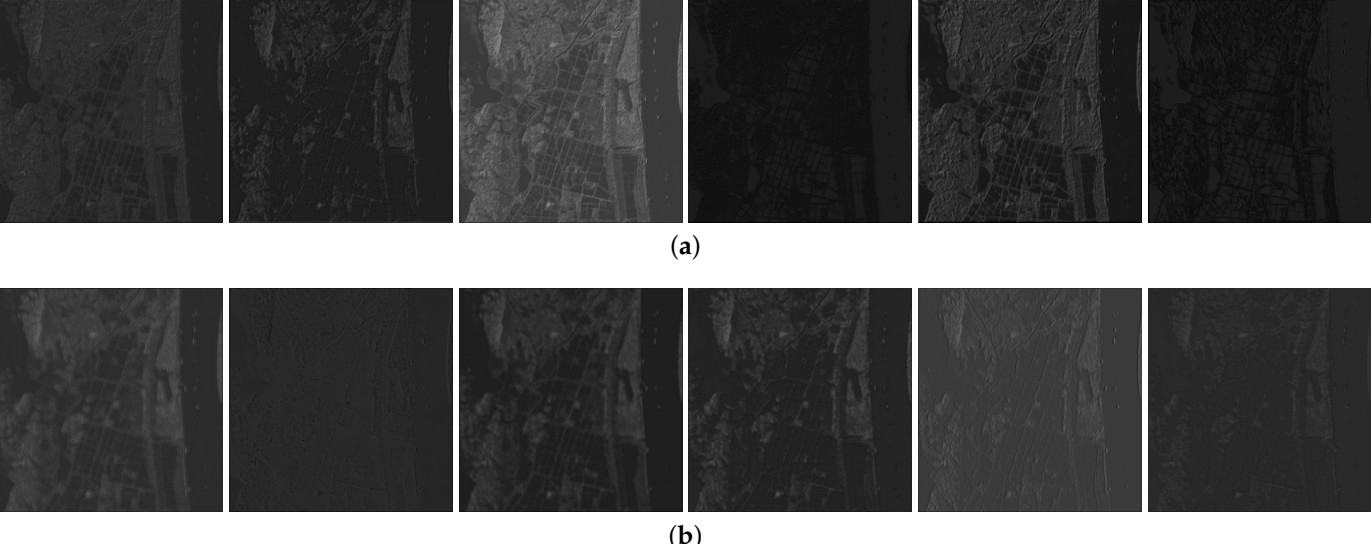

(**a**)

(**b**)

**Figure 5.** Example of the features extracted by two different modules. (**a**) Feature maps of MSFEB, (**b**) feature maps of basic convolutional module.

### 3.2. Spectral Retention Branch

The purpose of the spectral retention branch is to retain more spectral information of MS images when upsampling MS images and also to retain the target information in MS images. In this paper, the upsampling and resblock method is designed to achieve the upsampling of MS images. As shown in Figure 3, a residual block is introduced to extract the feature information of MS images after the upsampling operation using the bicubic linear interpolation operator. At the same time, it can eliminate the checkerboard artifacts generated by the upsampling operation.

The feature map obtained by spectral retention branch is shown in Figure 6. From the feature map, it can be seen that the spectral preservation branch preserves the spatial information of the MS image, and the differences between channels can also reflect the spectral information of the feature map.

### 3.3. Attention Feature Fusion Module

The feature maps of SAR and MS images are obtained after the spatial feature extraction branch and the spectral retention branch. In order to fully utilize the spatial information of the SAR image and the spectral information of the MS image, the feature maps must be fused. Unlike other networks that simply add the features of two images directly, the proposed attention feature fusion (AFF) module can calculate the weights

of each feature map by channel. Then let the feature maps be summed by channel with different weights. The structure of AFF module is shown in Figure 7. AFF model can be expressed as

$$OUT = M_1(X_1) \otimes X_1 \oplus M_2(X_2) \otimes X_2 \qquad (10)$$

where $X_1, X_2$ denotes the two input feature maps, $OUT$ denotes the output of AFF module, $M(X)$ denotes the adaptive weights obtained from the channel attention module, $\otimes$ is elemental multiplication, and $\oplus$ is summation by elements.

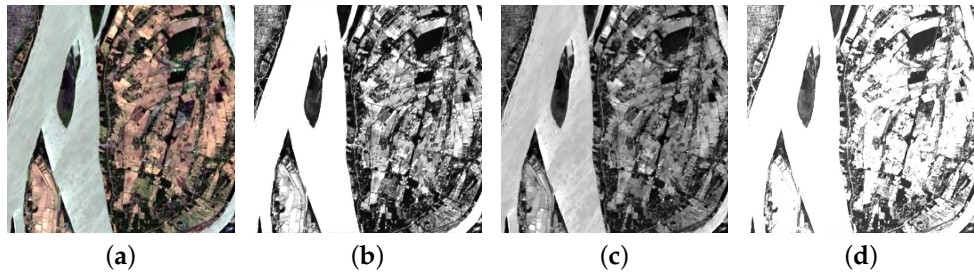

| (a) | (b) | (c) | (d) |

**Figure 6.** Example of the feature maps extracted by spectral retention branch. (**a**) Input MS image. (**b**) First feature map. (**c**) Second feature map. (**d**) Third feature map.

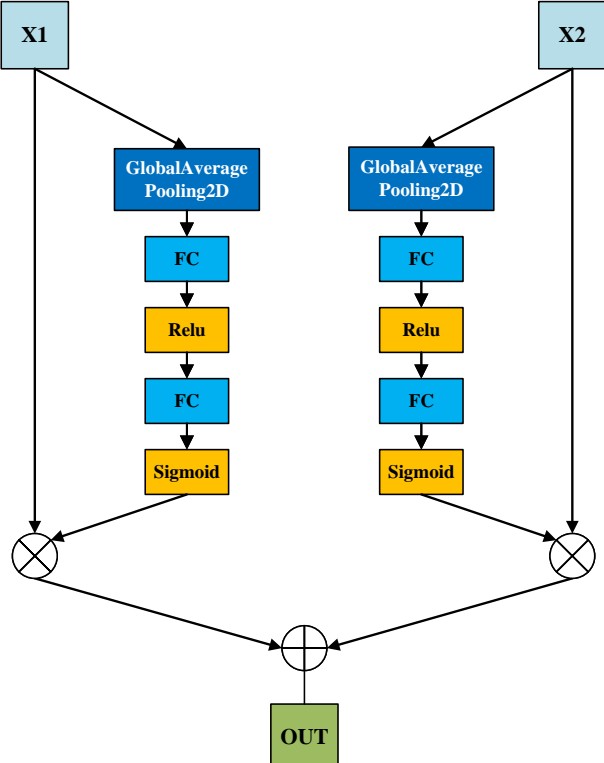

**Figure 7.** AFF module.

*3.4. Unsupervised Union Loss Function*

An unsupervised learning joint loss function is designed to train the network for the imaging mechanism of SAR images and MS images, the difficulty of obtaining real reference images, and the need to simultaneously consider the spatial detail information of SAR images and the spectral information of MS images. The union loss function contains spectral loss $L_{spectral}$ and detail loss $L_{spatial}$, which are calculated as

$$Loss = L_{spectral} + \lambda L_{spatial} \qquad (11)$$

where $\lambda$ denotes the weighting coefficient. The spectral loss function is expressed as the L1 norm between the fusion image and the multispectral image, which is used to constrain the spectral distortion of the fused image. The formula is defined as

$$L_{spectral} = |F - MS\uparrow|_1 \tag{12}$$

where $F$ denotes the fused image, $MS\uparrow$ is the 3-fold upsampled MS input image, and $|\cdot|_1$ denotes the L1 norm. The detail loss function represents the structural similarity between the fused image and the input SAR image, so that the fused image carries more spatial detail information of the SAR image. The formula is defined as

$$L_{spatial} = 1 - SSIM(F, SAR_{HP}) \tag{13}$$

where $SAR_{HP}$ is the high-pass filtered SAR image. The $SSIM$ calculation formula is shown in Equation (9).

## 4. Experiments and Results

### 4.1. Datasets

The training datasets in this paper are selected from Sentinel-1 and LandSat-8. Sentinel-1 is Global Monitoring for Environment and Security (GMES) of the European Space Agency, consisting of two satellites carrying a C-band synthetic aperture radar that provides continuous images (day, night, and various weather conditions). Landsat is a series of U.S. Earth observation satellite systems used to explore the Earth's resources and environment, mainly for resource exploration, environmental monitoring, natural disaster prevention, etc. In this paper, the Ground Range Detected (GRD) level SAR data of Sentinel-I with a resolution of 10 m and polarization of VH are selected as SAR datasets. The MS images are selected from Landsat-8, with 4, 3, and 2 bands consisting of true color images with a resolution of 30 m. The images contain features such as wide sea area, cities, vegetation, mountains, etc., which can verify the applicability of the proposed method. Two grouped images of the dataset are shown in Figure 8.

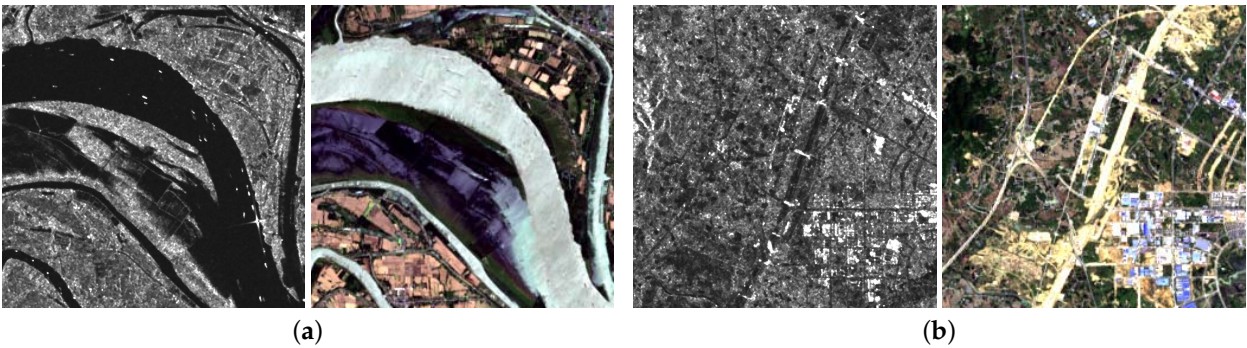

(**a**) (**b**)

**Figure 8.** Two groups images of Data set. (**a**) the first group, (**b**) the second group.

For more efficient training of the network, the source images are cropped into small patches.The MS images are cropped to a size of 32 × 32 pixels and the SAR images are cropped to a size of 96 × 96 pixels. A total of 25,600 pairs of images are cropped for training the network. In addition, a total of 656 pairs of SAR images and MS images of 768 × 768 and 256 × 256 pixel size are selected to test the network.

### 4.2. Experimental Setting

We split the dataset in the ratio of 9:1 as training and validation sets to train the network in 200 epochs. Batch size was set to 8, and the loss function was constrained using adaptive moment estimation (Adam). The initial learning rate was 0.0002 and the learning decay rate was 0.5. When the performance of the model did not improve after

three epochs of iteration, the learning rate reduction mechanism is triggered, and when the performance of the model does not improve after 10 epochs of iteration, the training process is terminated to avoid overfitting.

To further validate the effectiveness of the proposed union loss function, we set different values to train the network. Figure 9 shows the peak signal-to-noise ratio (PSNR) [43] (larger is better, Figure 9a) and spectral angle mapper (SAM) [44] (smaller is better, Figure 9b) on the test dataset after training the network with different lambda values. It can be seen from Figure 9 that PSNR decreases with the increase in value, whereas SAM increases with the increase in value. The best performance was obtained when $\lambda$ was 0.1. Therefore, in subsequent experiments, the value of $\lambda$ was set to 0.1. The network is implemented under python 3.8 and tensorflow 2.7. The experiments were performed on an NVIDIA (Santa Clara, CA, USA) GeForce RTX 3060 Laptop GPU.

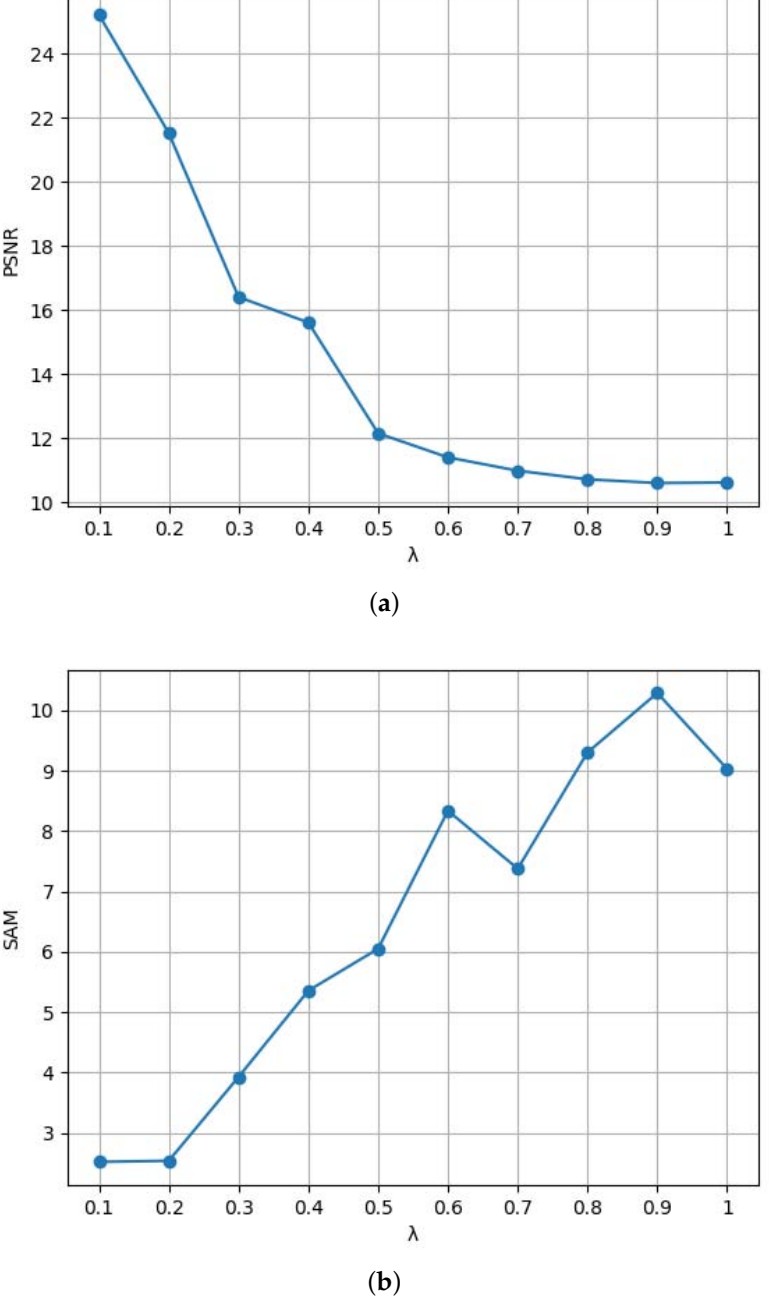

(**a**)

(**b**)

**Figure 9.** Result of $\lambda$ experimental. (**a**) PSNR, (**b**) SAM.

### 4.3. Comparison of Methods

In this paper, the proposed method is compared with six methods in terms of both visual performance and quantitative metrics.

(1)  IHS [6]: a fast intensity–hue–saturation fusion technique;
(2)  NSCT [27]: non-subsampled contourlet transform domain fusion method;
(3)  Wavelet [17]: the wavelet transform fusion method;
(4)  NSCT-FL [26]: a fusion method based on NSCT and fuzzy logic;
(5)  NSCR-PCNN [25]: a fusion method Based on NSCT and pulse-coupled neural network;
(6)  MSDCNN [35]: a multiscale and multidepth convolutional neural network. The MSDCNN is trained to constrain the training by using the loss function proposed in this paper.

### 4.4. Evaluation Indicators

In this paper, seven widely using metrics are selected to quantitatively evaluate the performance of the proposed method and the comparison method. Four of them are included as reference image evaluation metrics: correlation coefficient (CC) [45], peak signal-to-noise ratio (PSNR) [43], spectral angle mapper (SAM) [44], SSIM [42]. Three non-reference evaluation metrics are used as well, which include quality with no reference (QNR), spatial distortion ($D_s$) and spectral distortion ($D_\lambda$) [45]. In calculating the evaluation metrics with reference images, the triple upsampling MS images are used as reference images according to Wald's protocol [46].

The correlation coefficient (CC) reflects the correlation degree between the fused image and the reference image, and its value ranges from $[-1,1]$, and larger values indicate a higher correlation between the two images. The CC is defined as

$$CC = \frac{\sum\limits_{i}^{M}\sum\limits_{j}^{N}[(R(i,j) - \mu_R)(F(i,j) - \mu_F)]}{\sqrt{\sum\limits_{i}^{M}\sum\limits_{j}^{N}(R(i,j) - \mu_R)^2 \cdot \sum\limits_{i}^{M}\sum\limits_{j}^{N}(F(i,j) - \mu_F)^2}} \tag{14}$$

$$\mu_R = \frac{1}{M \times N}\sum\limits_{i}^{M}\sum\limits_{j}^{N}R(i,j) \tag{15}$$

$$\mu_F = \frac{1}{M \times N}\sum\limits_{i}^{M}\sum\limits_{j}^{N}F(i,j) \tag{16}$$

where $R, F$ are the reference image and the fused image (the following $R, F$ all represent this meaning), $\mu$ is the average value of the image, and $M, N$ are the height and width of the image.

The peak signal-to-noise ratio (PSNR) reflects the quality of the fused image by calculating the ratio between the maximum peak value of the fused image and the mean square error of the reference image. The PSNR is defined as

$$PSNR = 10\lg\frac{(MAX_I)^2}{\frac{1}{M \times N}\sum\limits_{i}^{M}\sum\limits_{j}^{N}(F(i,j) - R(i,j))^2} \tag{17}$$

where $MAX_I$ indicates the maximum value of image pixel points. A higher PSNR value between two images indicates that the two images are more similar.

A spectral angle mapper (SAM) is used to measure the degree of spectral distortion between the fused image and the corresponding pixel points of the reference image. The SAM is defined as

$$SAM = \arccos\left(\frac{\langle I_R, I_F \rangle}{\|I_R\| \cdot \|I_F\|}\right) \tag{18}$$

where $I_R, I_F$ is the vector at the same pixel point of the reference image and the fused image, $\langle \cdot, \cdot \rangle$ denotes the inner product of the two vectors, and $\|\cdot\|$ denotes the L2 norm. The ideal value of SAM is 0.

The structural similarity (SSIM) defines structural information from the perspective of image composition as a property that reflects the structure of objects in a scene independently of luminance and contrast. Furthermore, the distortion of the image is expressed as the combination of three different factors of luminance, contrast, and structure. The expression is shown in Equation (9), and the coefficient $\alpha, \beta, \gamma$ in Equation (9) is set to 1.

The quality with no reference (QNR) index reflects the quality of the fused image by measuring the spatial distortion between the MS image, the SAR image, and the fused image, as well as the spectral distortion between the MS image and the fused image. The optimal value of QNR is 1, where the quality of the fused image is the highest. QNR is defined as

$$QNR = (1 - D_\lambda)^\alpha (1 - D_s)^\beta \tag{19}$$

where $\alpha, \beta$ usually is set to 1, $D_\lambda$ and $D_s$ are obtained by calculating the $Q$ [41] index between different images. The index Q [39] is commonly used to measure image distortion and is calculated as

$$Q = \frac{2\mu_R \mu_F}{\mu_R^2 + \mu_F^2} \frac{|\sigma_{RF}|}{\sigma_R \cdot \sigma_F} \frac{2\sigma_R \cdot \sigma_F}{\sigma_R^2 + \sigma_F^2} \tag{20}$$

$D_\lambda$ measures the spectral distortion of the image by calculating the $Q$ value between the respective channels of the fused image and the original MS image. The optimal value of $D_\lambda$ is 0. The $D_\lambda$ is calculated as

$$D_\lambda = \sqrt[p]{\frac{1}{L(L-1)} \sum_{l=1}^{L} \sum_{r=1, r \neq l}^{L} \left(Q(F^l, F^r) - Q(MS^l, MS^r)\right)^p} \tag{21}$$

where $MS$ denotes the original multispectral image, and $L$ denotes the number of channels of the fused image. p is usually set to 1.

$D_s$ measures the degree of spatial distortion of the fused image by calculating the difference between the $Q$ metric between the fused image and the SAR image channel and the $Q$ value between the original MS image and the degraded SAR image. The optimal value of $D_s$ is 0, and the $D_s$ is calculated as

$$D_s = \sqrt[q]{\frac{1}{L} \sum_{l=1}^{L} \left(Q(F^l, SAR) - Q(MS^l, SAR_{LR})\right)^q} \tag{22}$$

where $SAR$ denotes the input SAR image and $SAR_{LR}$ denotes the downsampled SAR image. $q$ is usually set to 1.

### 4.5. Analysis of Results

In the experiments, two groups of images from the test dataset of 656 images containing various types of features were selected to analyze the advantages and disadvantages between the proposed method and the comparison method in terms of visual effects and objective evaluation indexes. In the table of objective evaluation indexes, the best values are indicated in bold, the second best values are marked with underlined horizontal lines, and the third values are marked with sliding wavy lines.

The first group is the image with the river, and the fusion results of each method are shown in Figure 10. Figure 10a is the reference image, Figure 10b is the original SAR image, and Figure 10c–i show the fusion results of various methods. In order to be able to more effectively observe the details of the fusion images, the red and yellow boxed areas in the figure are enlarged as in Figures 11 and 12. As can be seen in Figure 10, the HIS, NSCT, and wavelet spectra are severely distorted and also exhibit severe spatial distortion. NSCT-FL and NSCT-PCNN have good performance in spectral retention, but exhibit severe spatial detail in the form of spatial distortion. In comparison, the CNN-based methods perform better in spectral retention compared to the conventional methods. However, the MSDCNN is inferior to the DAFCNN in terms of feature extraction capability, as shown in Figure 11. In the yellow enlarged area, it can be clearly seen from Figure 12 that the MSDCNN has some spectral distortion. From the objective evaluation results, as shown in Table 1, the performance of the CNN-based fusion method is far ahead of other methods, and DAFCNN is in the leading position.

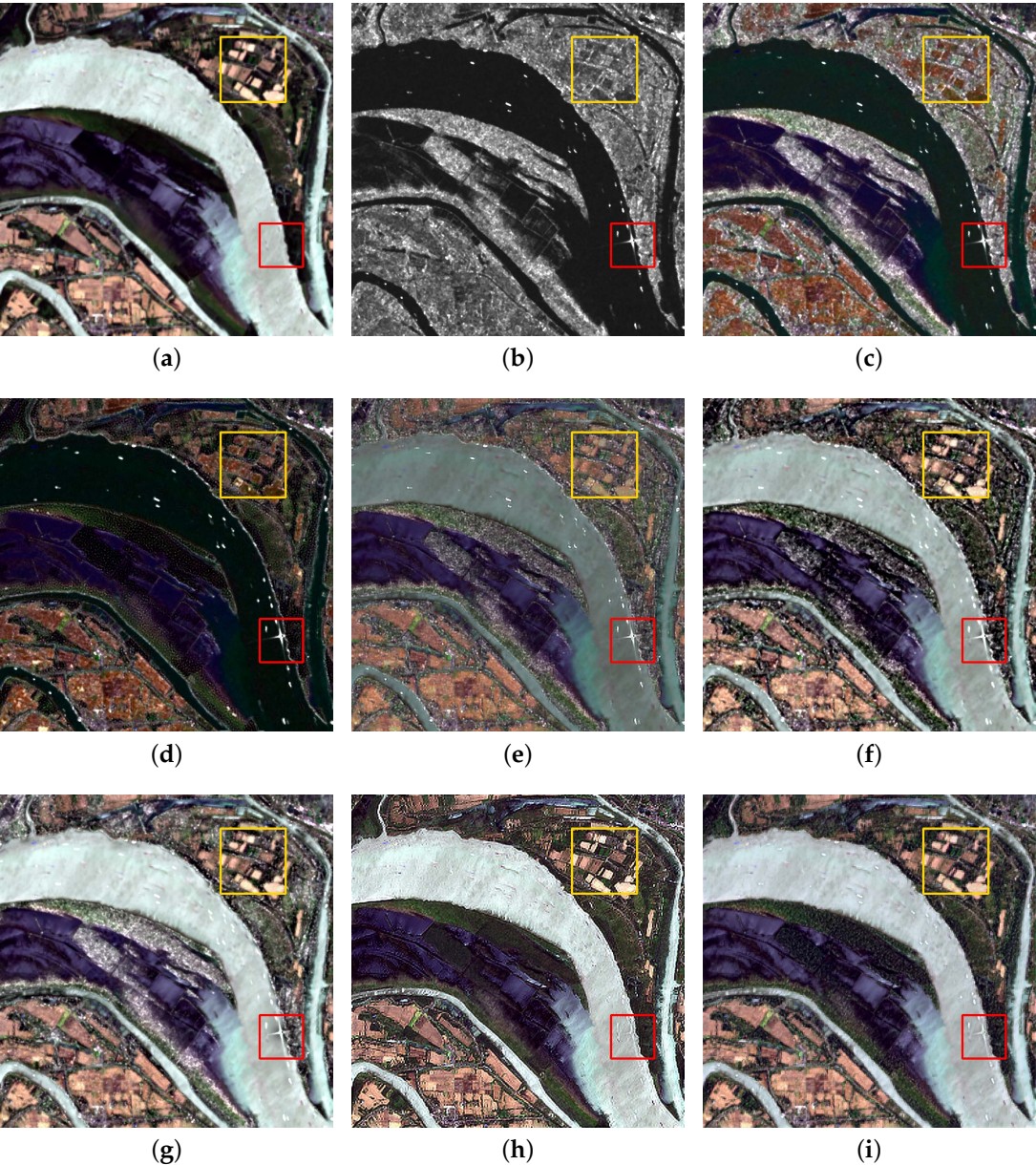

**Figure 10.** The fusion results of different methods for the first group of images. (**a**) Reference, (**b**) SAR, (**c**) IHS, (**d**) NSCT, (**e**) Wavelet, (**f**) NSCT-FL, (**g**) NSCT-PCNN, (**h**) MSDCNN, (**i**) DAFCNN.

**Table 1.** Quantitative indicators of the first group fusion results.

| Methods | CC (↑) | PSNR (↑) | SAM (↓) | SSIM (↑) | $D_s$ (↓) | $D_\lambda$ (↓) | QNR (↑) |
|---|---|---|---|---|---|---|---|
| IHS | −0.5820 | 6.3007 | 10.6878 | −0.4972 | 0.1155 | 0.0027 | 0.8821 |
| NSCT | −0.0512 | 6.9353 | 14.9826 | −0.0192 | 0.2648 | 0.0049 | 0.7315 |
| Wavelet | 0.6155 | 11.5314 | 3.1822 | 0.4338 | 0.0904 | 0.0017 | 0.9080 |
| NSCT-FL | 0.8535 | 15.6627 | **2.0313** | 0.8285 | 0.0215 | **0.0003** | 0.9782 |
| NSCT-PCNN | 0.8120 | 14.0786 | 2.3038 | 0.7873 | 0.1045 | 0.0011 | 0.8945 |
| MSDCNN | 0.9371 | 18.9213 | 2.6944 | 0.9337 | 0.0133 | 0.0004 | 0.9861 |
| DAFCNN | **0.9799** | **2.5795** | 2.4434 | **0.9679** | **0.0074** | 0.0006 | **0.9919** |

↑: The larger the value, the better. ↓: The smaller the value, the better. **bold format**: The best value. underline: The second best value. under wave lines: The third best value.

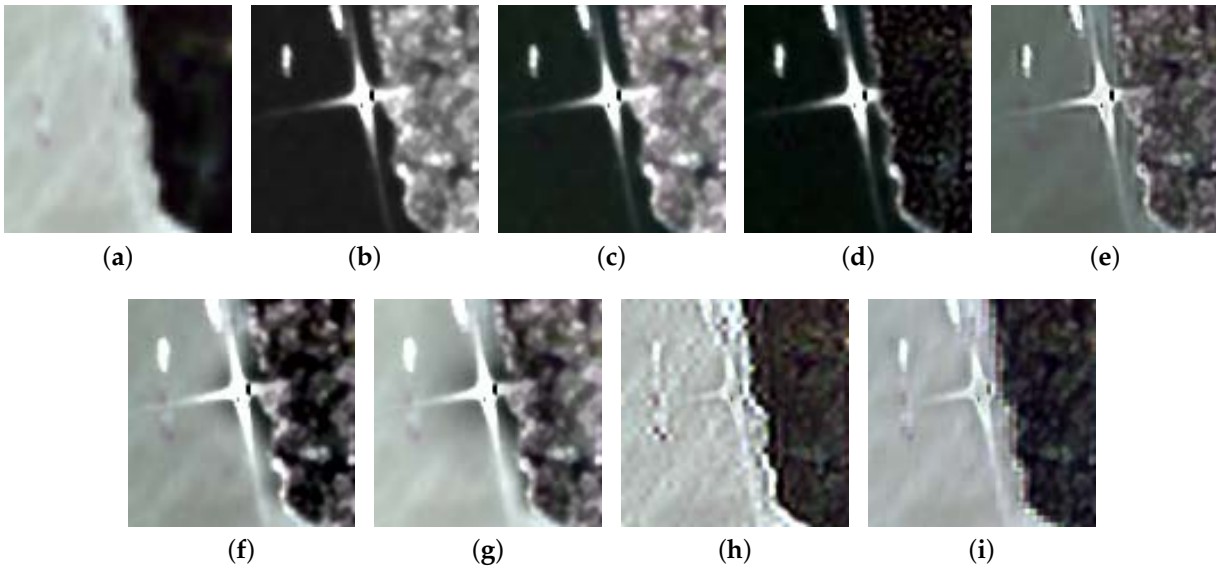

**Figure 11.** Enlarged red box area in Figure 10. (**a**) Reference, (**b**) SAR, (**c**) IHS, (**d**) NSCT, (**e**) Wavelet, (**f**) NSCT-FL, (**g**) NSCT-PCNN, (**h**) MSDCNN, (**i**) DAFCNN.

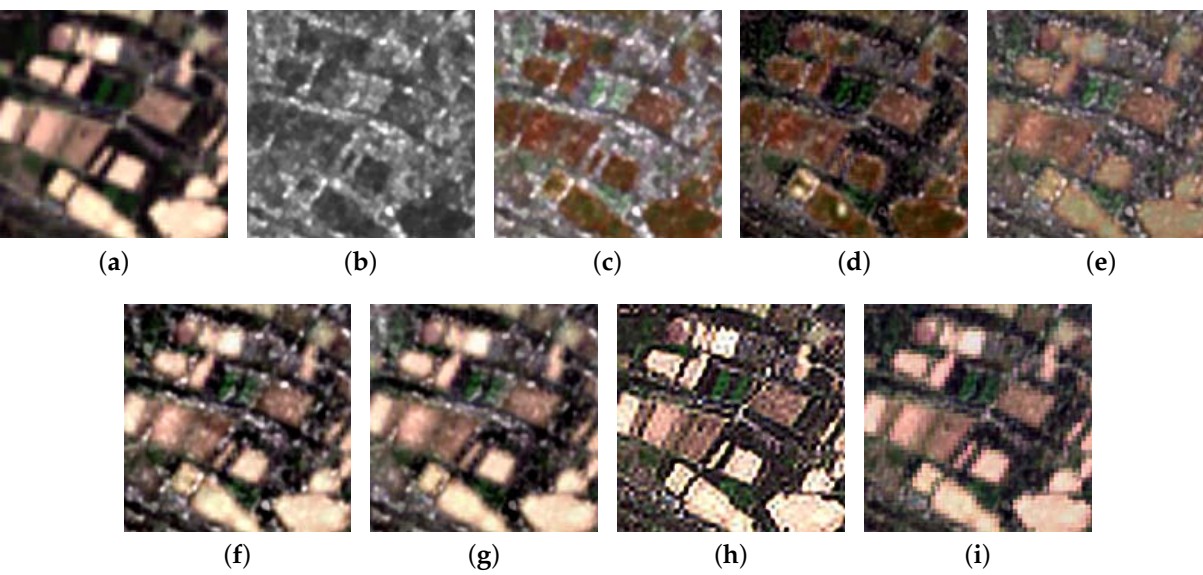

**Figure 12.** Enlarged yellow box area in Figure 10. (**a**) Reference, (**b**) SAR, (**c**) IHS, (**d**) NSCT, (**e**) Wavelet, (**f**) NSCT-FL, (**g**) NSCT-PCNN, (**h**) MSDCNN, (**i**) DAFCNN.

The fusion results of the second group images, which include urban and mountain areas are shown in Figure 13. Reference image and the original SAR image are shown in Figure 13a,b, and Figure 13c–i show the fusion results of various methods. Similarly, the red boxed area in the fusion results are enlarged as shown in Figure 14. from the fusion results it can be seen that the traditional methods have serious spectral and spatial distortion, and the CNN-based fusion method still outperforms the traditional method. However, as shown in Figure 14, the MSDCNN introduces too much speckle noise, which leads to distortion of the object structure in the image. the DAFCNN method adds more details while obtaining better spectral fidelity. The objective evaluation metrics corresponding to Figure 13 are shown in Table 2, and it can be seen that DAFCNN exhibits optimal fusion values for most metrics.

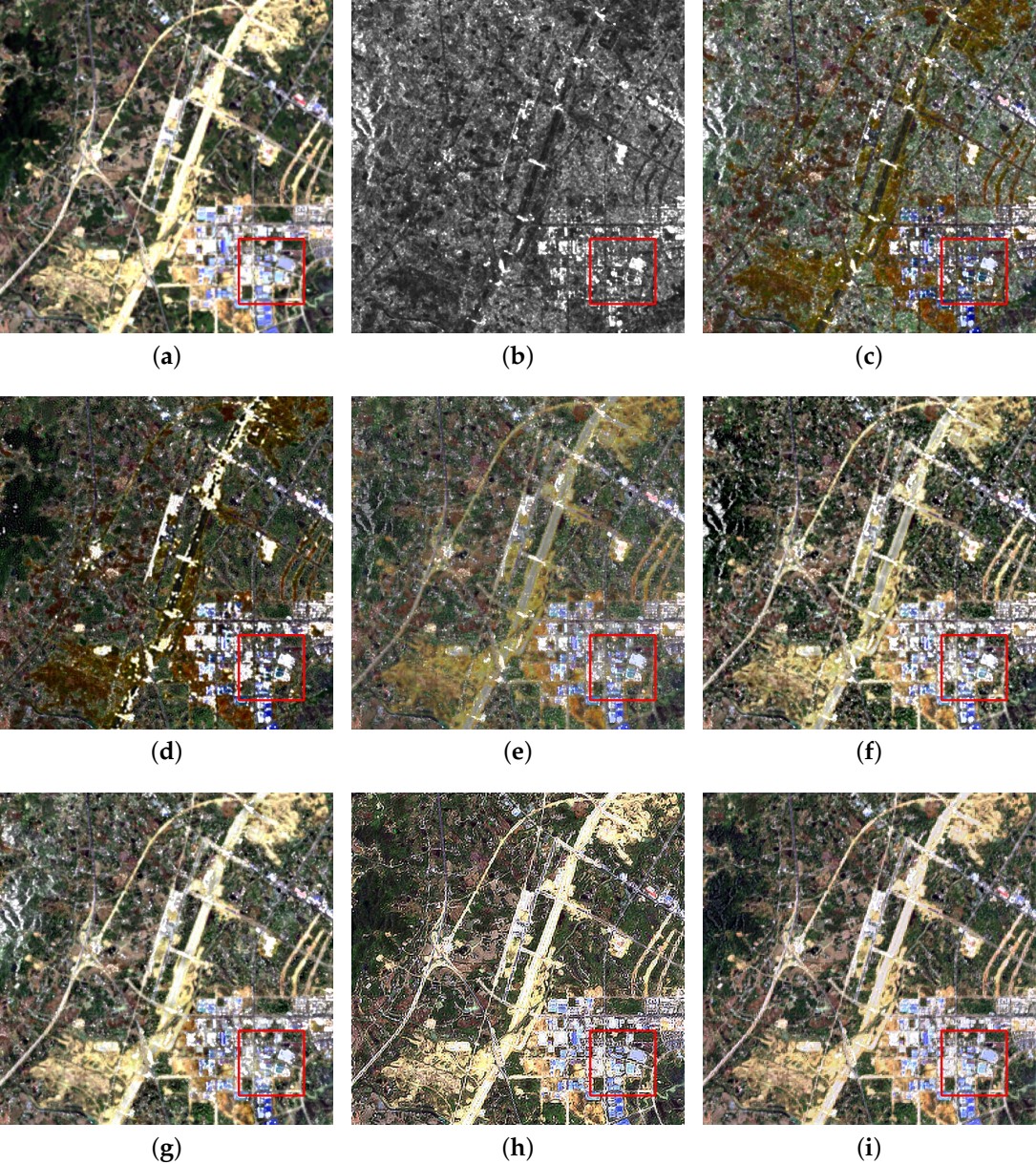

**Figure 13.** The fusion results of different methods for the second group of images. (**a**) Reference, (**b**) SAR, (**c**) IHS, (**d**) NSCT, (**e**) Wavelet, (**f**) NSCT-FL, (**g**) NSCT-PCNN, (**h**) MSDCNN, (**i**) DAFCNN.

**Table 2.** Quantitative indicators of the second group fusion results.

| Methods | CC (↑) | PSNR (↑) | SAM (↓) | SSIM (↑) | $D_s$ (↓) | $D_\lambda$ (↓) | QNR (↑) |
|---|---|---|---|---|---|---|---|
| IHS | −0.0123 | 9.5775 | 8.5049 | −0.4900 | 0.0390 | 0.0091 | 0.9522 |
| NSCT | 0.5055 | 10.7658 | 9.0350 | 0.4479 | 0.0670 | 0.0218 | 0.9121 |
| Wavelet | 0.7380 | 14.3651 | 3.6564 | 0.6610 | 0.0281 | 0.0063 | 0.9658 |
| NSCT-FL | 0.8493 | 16.5720 | 2.7567 | 0.8367 | 0.0135 | 0.0030 | 0.9835 |
| NSCT-PCNN | 0.8346 | 16.2621 | **2.4721** | 0.8189 | 0.0154 | **0.0013** | 0.9833 |
| MSDCNN | 0.8577 | 15.9557 | 4.6426 | 0.8442 | 0.0089 | 0.0018 | 0.9893 |
| DAFCNN | **0.9750** | **22.3246** | 3.4506 | **0.9565** | **0.0067** | 0.0026 | **0.9908** |

↑: The larger the value, the better. ↓: The smaller the value, the better. **bold format**: The best value. <u>underline</u>: The second best value. <u>under wave lines</u>: The third best value.

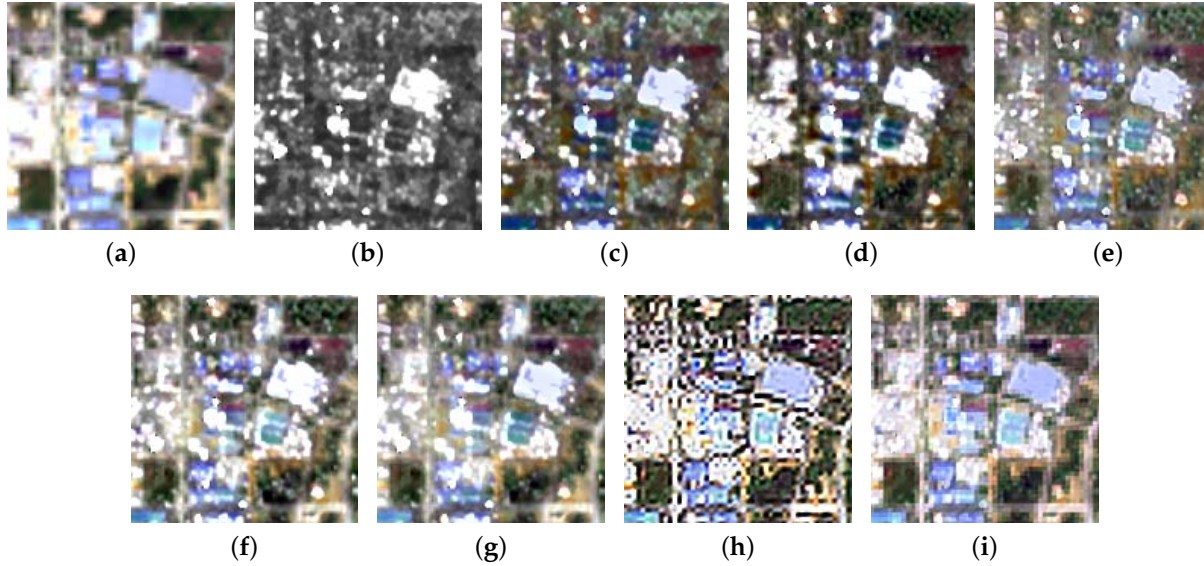

**Figure 14.** Enlarged red box area in Figure 13. (**a**) Reference, (**b**) SAR, (**c**) IHS, (**d**) NSCT, (**e**) Wavelet, (**f**) NSCT-FL, (**g**) NSCT-PCNN, (**h**) MSDCNN, (**i**) DAFCNN.

Table 3 shows the average values of objective evaluation metrics for the entire set containing 656 sets of images. From the table, it can be seen that the method DAFCNN performs optimally in most of the evaluation metrics, which also proves that our proposed method is effective for the task of fusing SAR images with MS images.

**Table 3.** Average quantitative metrics for the test dataset of 656 images.

| Methods | CC (↑) | PSNR (↑) | SAM (↓) | SSIM (↑) | $D_s$ (↓) | $D_\lambda$ (↓) | QNR (↑) |
|---|---|---|---|---|---|---|---|
| IHS | 0.2909 | 8.6195 | 15.3945 | 0.0362 | 0.3730 | 0.2945 | 0.4765 |
| NSCT | 0.2607 | 8.9561 | 20.6113 | 0.1845 | 0.2380 | 0.2714 | 0.5421 |
| Wavelet | 0.7017 | 13.2078 | 4.6937 | 0.5699 | 0.1986 | 0.0431 | 0.7685 |
| NSCT-FL | 0.7061 | 19.2868 | 2.1126 | 0.6904 | 0.0296 | 0.0056 | 0.9649 |
| NSCT-PCNN | 0.6940 | 20.0053 | **2.0886** | 0.6740 | 0.0504 | **0.0046** | 0.9472 |
| MSDCNN | 0.8913 | 21.1186 | 3.0055 | 0.8466 | 0.0423 | 0.0053 | 0.9528 |
| DAFCNN | **0.9801** | **25.2072** | 2.5177 | **0.9394** | **0.0230** | 0.0053 | **0.9718** |

↑: The larger the value, the better. ↓: The smaller the value, the better. **bold format**: The best value. <u>underline</u>: The second best value. <u>under wave lines</u>: The third best value.

### 4.6. Validation of the Performance of the Proposed Fusion Module

When fusing feature maps, CNN-based fusion methods usually directly add two feature maps in a simple linear manner without considering the relationship between different feature maps. This fusion strategy may not yield the desired fusion results. In

order to fully utilize the connections between feature map channels, the AFF module shown in Figure 7 has been designed. The AFF module adds adaptive weights to the fused feature maps through the attention mechanism.

To verify the effectiveness of the AFF module, the fusion strategy of feature maps summed by elements is used in place of AFF module in the DAFCNN. The new fusion network is called DAFCNN-ADD, and the original is called DAFCNN-AFF. As shown in Figure 15, the DAFCNN-AFF has superior ability in spectral retention. the results of DAFCNN-AFF are closer to the reference image. From this, it can be seen that adding the AFF module significantly improves the performance of the network. Furthermore, as shown in Table 4, Whether in reference image evaluation indicators or no reference image evaluation indicators, the DAFCNN-AFF outperforms the DAFCNN-ADD in almost all quantitative metrics. This also indicates that the proposed AFF module is useful for the fusion of SAR images and MS images.

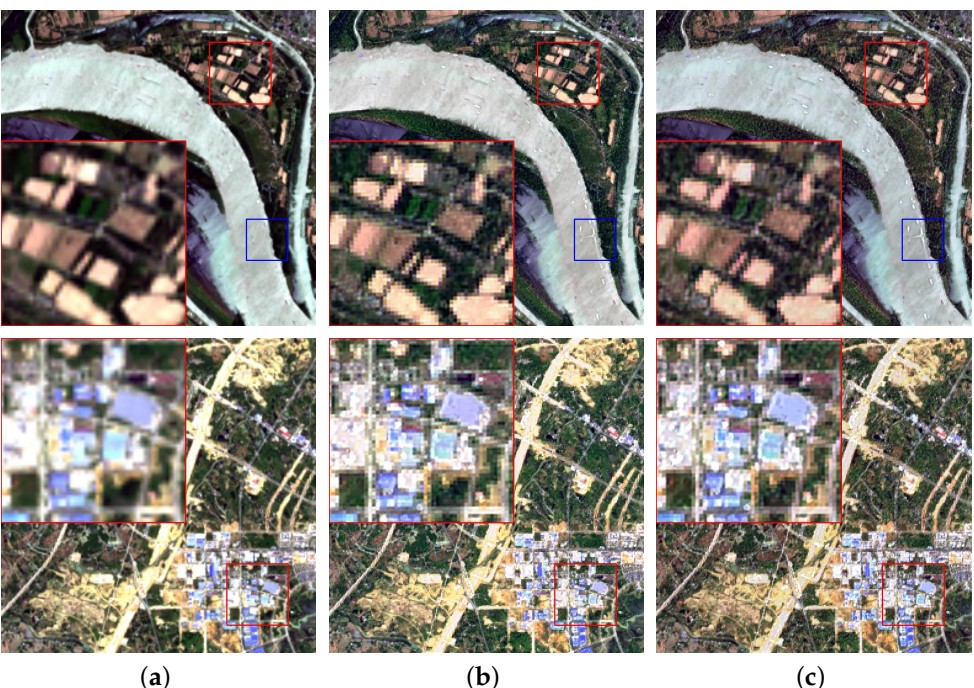

**Figure 15.** Verification of the performance of the AFF module. (**a**) Reference, (**b**) DAFCNN-ADD, (**c**) DAFCNN-AFF.

**Table 4.** Verification of the performance of the AFF module.

| Methods | CC (↑) | PSNR (↑) | SAM (↓) | SSIM (↑) | $D_s$ (↓) | $D_\lambda$ (↓) | QNR (↑) |
|---|---|---|---|---|---|---|---|
| DAFCNN-AFF | **0.9801** | **25.2072** | **2.5177** | 0.9394 | **0.0230** | **0.0053** | **0.9718** |
| DAFCNN-ADD | 0.9645 | 24.4646 | 2.7464 | **0.9395** | 0.0401 | 0.0108 | 0.9459 |

↑: The larger the value, the better. ↓: The smaller the value, the better. **bold format**: The better value.

## 5. Conclusions

In this paper, a dual-channel feature extraction and attention fusion convolutional neural network (DAFCNN) is proposed to implement SAR image and MS image fusion. In this experiment, the high-frequency components of SAR images are mainly used to obtain spatial information of the images. In order to extract the features of high-resolution SAR images more effectively, a dual-channel feature extraction module is designed to extract the spatial feature information of high-resolution SAR images. The residual block is introduced after the feature extraction module, which not only can obtain deeper feature maps but also can reduce the information loss caused by the deeper network. Finally, unlike the common

simple direct summation fusion strategy, an attention-based feature fusion (AFF) module is designed, which takes fully into account the relationship between feature map channels and shows excellent ability in spectral preservation. An upsampling+resblock structure is used in the experiments to achieve upsampling of low-resolution MS images, which can not only eliminate the tessellation effect but also better utilize the spatial information in MS images. Finally, the unsupervised union loss function is used in the training phase to constrain the network learning. The experimental results show that the proposed method has excellent performance in the task of fusing SAR images with MS images. The fusion of MS images and SAR images generates high-quality fused images with richer spatial information and spectral features. It is also of great significance for urban land cover, terrain classification, road detection, and other tasks. In the future, more methods will be sought to achieve SAR image and MS image fusion instead of focusing on CNN only, such as by GAN, GCN, and other methods to achieve SAR image and MS image fusion.

**Author Contributions:** Conceptualization, F.Z. and M.X.; methodology, J.L. and F.Z.; validation, F.Z. and J.L.; formal analysis, J.L.; investigation, J.Y.; data curation, F.Z. and J.Y.; writing—original draft preparation, J.L.; writing—review and editing, F.Z. and J.L.; supervision, M.X.; project administration, F.Z.; funding acquisition, F.Z. All authors have read and agreed to the published version of the manuscript.

**Funding:** This work was supported in part by the Natural Science Foundation of Anhui Province under grant 2208085MF156; in part by the Fundamental Research Funds for the Central Universities under grant JZ2022HGTB0329; and in part by the Aeronautical Science Foundation of China under grant 2019200P4001.

**Data Availability Statement:** Not applicable.

**Conflicts of Interest:** The authors declare no conflict of interest.

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
