# Peer review of "DAFCNN: A Dual-Channel Feature Extraction and Attention Feature Fusion Convolution Neural Network for SAR Image and MS Image Fusion"

_remotesensing, doi:10.3390/rs15123091_

Round 1

Reviewer 1 Report

1. SENet is used in this paper. Compared with other attention mechanisms, what are the advantages of SENet?

2. The explanations of the unknowns in Equations 3, 4 and 5 are incomplete.

3. 3.2. It is suggested to add some graphs in the Spectral Retention Branch to prove the effectiveness of this method.

4. Explain why the data set used in this paper is suitable for this model.

5. 4.3. IHS, NSCT, etc. in the comparison method are suggested to use the serial number of the lower level.

6. In the summary, the author should discuss these findings and their impact. And emphasize the future research direction.

7. Most of the cited references are too old to show the advanced nature of the paper.

Minor editing of English language required

Reviewer 2 Report

This paper presents a MS/SAR image fusion method based on neutral networks. There is some novelty in the design of the network architecture and the technique overall sounds good to this reviewer. I therefore recommend publication of this work, subject to addressing some minor comments below. 

1.  There are several places in this paper where an abbreviation is introduced but only a few paragraphs after their full spelling is given. However, it is a general convention to give the full spelling of a term the first time an abbreviation is introduced. 

2. Page 10, line 250, can you also explain what is the Wald's protocol? And why based on it the MS images are used as reference?

3. I am not super convinced why the PSNR is a good metric. In particular, clearly the PSNR is higher if MAX_I is higher, but why a high MAX_I is a good thing? 

Some minor grammatical improvements seem needed. But the quality overall looks good and does not present proper understanding.

Reviewer 3 Report

The authors of the article have created a method that is proven by the examples given to be very efficient and worthy of consideration in future research.

In this paper presented by the authors of the study, the Sentinel 1 SAR image and the Landsat 8 MS image are used as datasets for the experiments.

In this paper, a neural network (DAFCNN) is proposed to implement SAR image and MS image fusion.

This experiment, the high-frequency components of SAR images are mainly used to obtain the spatial information of the images.

In order to extract the features of high-resolution SAR images more efficiently, a two-channel feature extraction module is designed to extract spatial feature information of high-resolution SAR images.

The authors of the article introduced the residual block after the feature extraction module, which can not only obtain deeper feature maps, but also reduce the information loss caused by the deeper network.

  The experimental results show that the proposed method has excellent performance in the task of fusing SAR images with MS images.

In the future, the authors have as research direction, more methods will be sought to achieve the fusion of SAR image and MS image instead of focusing only on CNN, such as through GAN, GCN and other methods to achieve SAR image and fusion of the MS image.

The paper proposes a fusion method of the SAR image and the MS image based on the convolutional neural network and is a new approach to solving the problem.

In order to use the spatial information and different scale information of the high-resolution SAR image, the authors of the article built a two-channel feature extraction module to obtain the feature map of the SAR image.

Experimental results show that, compared to traditional fusion methods and deep learning algorithms, the proposed algorithm has better performance in quantitative and visual representation and is clearly superior to other previous methods and approaches.

The authors of the article have created a method that is proven by the examples given to be very efficient and worthy of consideration in future research.

In this paper presented by the authors of the study, the Sentinel 1 SAR image and the Landsat 8 MS image are used as datasets for the experiments.

In this paper, a neural network (DAFCNN) is proposed to implement SAR image and MS image fusion.

This experiment, the high-frequency components of SAR images are mainly used to obtain the spatial information of the images.

In order to extract the features of high-resolution SAR images more efficiently, a two-channel feature extraction module is designed to extract spatial feature information of high-resolution SAR images.

The authors of the article introduced the residual block after the feature extraction module, which can not only obtain deeper feature maps, but also reduce the information loss caused by the deeper network.

  The experimental results show that the proposed method has excellent performance in the task of fusing SAR images with MS images.

In the future, the authors have as research direction, more methods will be sought to achieve the fusion of SAR image and MS image instead of focusing only on CNN, such as through GAN, GCN and other methods to achieve SAR image and fusion of the MS image.

The paper proposes a fusion method of the SAR image and the MS image based on the convolutional neural network and is a new approach to solving the problem.

In order to use the spatial information and different scale information of the high-resolution SAR image, the authors of the article built a two-channel feature extraction module to obtain the feature map of the SAR image.

Experimental results show that, compared to traditional fusion methods and deep learning algorithms, the proposed algorithm has better performance in quantitative and visual representation and is clearly superior to other previous methods and approaches.
